# Characterization of the Chloroplast Genome Structure of *Gueldenstaedtia verna* (Papilionoideae) and Comparative Analyses among IRLC Species

**Ogyeong Son [1] and Kyoung Su Choi [2],***

[1] Educational Research Division, Exhibition Research Department, Daegu National Science Museum, Yugaeup, Daegu 43023, Republic of Korea
[2] Plant Research Team, Animal and Plant Research Department, Nakdonggang National Institute of Biological Resources, Sangju 37242, Republic of Korea
* Correspondence: choiks010@nnibr.re.kr

**Abstract:** The genus *Gueldenstaedtia* belongs to Papilionaceae's inverted repeat-lacking clade (IRLC) and includes four species distributed throughout Asia. We sequenced the chloroplast genome of *G. verna* and compared it with those of the IRLC clade. The genome was 122,569 bp long, containing 77 protein-coding genes, 30 tRNAs, and 4 rRNAs. Comparative analyses showed that *G. verna* lost one inverted repeat region, the *rps16* gene, an intron of *rpoC1*, and two introns of *clpP*. Additionally, *G. verna* had four inversions (~50 kb inversion, *trnK–psbK*; ~28 kb inversion, *accD–rpl23*; ~10 kb inversion, *rps15–trnL*; ~6 kb inversion, *trnL–trnI*) and one reposition (*ycf1*). Its G + C content was higher than that of other IRLC species. The total length and number of repeats of *G. verna* were not significantly different from those of the other IRLC species. Phylogenetic analyses showed that *G. verna* was closely related to *Tibetia*. A comparison of substitution rates showed that *ycf2* and *rps7* were higher than one, suggesting that these were positive selection genes, while others were related to purified selection. This study reports the structure of the chloroplast genome of a different type, i.e., with four inversions and one reposition, and would be helpful for future research on the evolution of the genome structure of the IRLC.

**Keywords:** *Gueldenstaedtia verna*; IRLC; gene and intron loss; chloroplast genome structure; comparative analysis

## 1. Introduction

The chloroplast (cp) genomes of angiosperms have been used for phylogenetic analysis [1,2], nucleotide substitution analysis [3,4], cp genome evolution analysis [5,6], and DNA molecular marker analysis [7,8] over the past decades. Previous studies have demonstrated that the cp genome structure of angiosperms comprises large single-copy regions, small single-copy regions, and two inverted repeat (IR) regions [9,10]. Gene content and order are highly conserved, including 79 protein-coding genes, 29 tRNA genes, and 4 rRNA genes [10]. However, some angiosperms show differences in gene content, order, and structure. For example, members of Fabaceae [11], Geraniaceae [3,12], Companulaceae [13–15], and Orobanchaceae [16–18] showed rearrangement of gene order, inversion, loss in IR regions, expansion of IR regions, loss of genes, or pseudogenes.

Fabaceae is one of the largest families of angiosperms containing essential species for agricultural activities. Fabaceae is classified into six subfamilies: Caesalpinioideae, Cercidoideae, Detarioideae, Dialioideae, Duparquetioideae, and Faboideae (Papilionoideae) [19]. Among the subfamilies, Faboideae, one of its monophyletic clades, known as the inverted repeat lacking clade (IRLC), has lost one copy of the IR region (25 kb) in the cp genome [20]. The IRLC includes 52 genera and over 4000 species divided into seven tribes, and the cp genomes of IRLC species show a loss or pseudogenization of genes (*rps16*, *rpl22*, *infA*,

*accD*, and *ycf4*), loss of introns (*clpP*, *atpF*, and *rpoC1*), inversions, and gene transfer to the nucleus [20–24]. Recently, Choi et al. [25] suggested the IR re-emergence in one IRLC species, *Medicago minima*.

*Gueldenstaedtia* is a genus of papilionoid legumes established by Fischer and named after Gueldenstaedt [26]. Sanderson and Wojciechowski's [27] molecular analysis included *G. himalaica* under the *Astragalus* genus due to its close relation with *Chesneya dshungarica*, although it was supported by low bootstrap values (30%). Later, Zhu [28] suggested dividing the genus *Gueldenstaedtia* into two subgenera, *Gueldenstaedtia* and *Tibetia,* once the two groups were distinct in seeds and other morphological traits, pollen characteristics, and chromosome data [28,29]. Only four species of the genus *Gueldenstaedtia* (*G. monophylla*, *G. thihangensis*, *G. henryi,* and *G. verna*) are distributed throughout Asia [29]. Recently, molecular phylogenetic analyses using a nuclear internal transcribed spacer (ITS) and plastids *matK*, *trnL-F*, and *psbA-trnH* showed that *Gueldenstaedtia* and *Tibetia* (GUT clade) are closely related, being supported by the highest bootstrap value (100%). Both analyses also placed *Chesneya* as a sister clade to GUT [30]. In previous studies, some of the cp genomes in IRLC species independently showed genomic rearrangements, such as intron loss and gain, pseudogenization, and inversions [23,31]. There are reports of cp genome analyses of *Tibetia* species (*T. himalaya*, NC_053369 and *T. liangshanensis*, NC_036109), but the cp genome of *Gueldenstaedtia* has never been analyzed.

In this study, we report the novel and complete cp genome of *G. verna* in Fabaceae. We aimed to (1) compare the cp genomes within Fabaceae considering inversion, gene, and intron loss; (2) suggest a new phylogenetic position for the genus *Gueldenstaedtia*; and (3) determine the nucleotide substitution rates of *G. verna*.

## 2. Results

### 2.1. Characterization of the Chloroplast Genome of Gieldenstaedtia verna

A total of 32,505,084 reads were obtained after whole-genome sequencing (Figure S1). The size of the cp genome of *G. verna* (Genbank accession number: OP525440) was 122,569 bp, and it showed an IR loss (Figure 1). The GC content was 36.0%, and the total genes included 77 protein-coding genes (PCGs), 30 transfer RNA (tRNA), and 4 ribosomal RNA (rRNA). Among these genes, *Rps16* has been lost in *G. verna*, thirteen genes (*atpF*, *ndhA*, *ndhB*, *petB*, *petD*, *rpl2*, *rpl16*, *rps12*, *trnG*-UCC, *trnL*-UAA, *trnV*-UAC, *trnI*-GAU, and *trnA*-UGC) contained a single intron, and one gene (*ycf3*) contained two introns.

### 2.2. Comparison of cp Genomes within Fabaceae

The total length of IRLC species ranged from 121,020 bp (*Lathyrus sativus*) to 131,179 bp (*Wisteria brachybotrys*), and the GC content ranged from 33.8% (*Medicago hybrida*) to 36.0% (*G. verna*) (Table 1).

A few genes in the cp genomes of IRLC species were lost. The *rps16* gene was lost in all IRLC species. Genes *rps18*, *rpl23*, *atpE*, and *ycf4* were lost independently (Table 2). Only the *rps16* gene was lost in *G. verna*. The intron content in IRLC cp genomes was more variable than the gene content. The intron of *atpF* was lost in *L. frutescens*, and the intron of *rpl16* was lost *in T. aureum*. The intron of *rpoC1* was lost in *G. verna*. *L. japonicus* had two introns for *clpP*, whereas IRLC species lost one or two introns for *clpP*. Two *Tibetia* species (*T. himalacia* and *T. liangsharensis*), *G. verna,* and *Glycyrrhiza lepidota* did not have introns for *clpP* (Figure 2, Table 2). Five tRNA genes containing one intron were identified. However, the intron of *trnG*-UCC lost in four species (*M. hybrid*, *T. aureum*, *L. culinaris*, and *V. sativa*).

Among sequences longer than 30 bp, the repetitive sequence analysis detected sequences with 35 (*Lessertia frutescens*) to 236 (*Tibetia himalaica*) repeats. The length of the repeats varied between 30 bp and 517 bp (*V. sativa*), mostly forward repeats, except in *L. frutescens* and *Lotus japonicas* (not IRLC species) (Table 1). The abundance of repetitive sequences in seven species, including *L. japonicus* (not an IRLC species), was below 3%, whereas another seven species had around 3% or slightly more. *T. himalaica* had the highest percentage of repetitive sequences (9.5%).

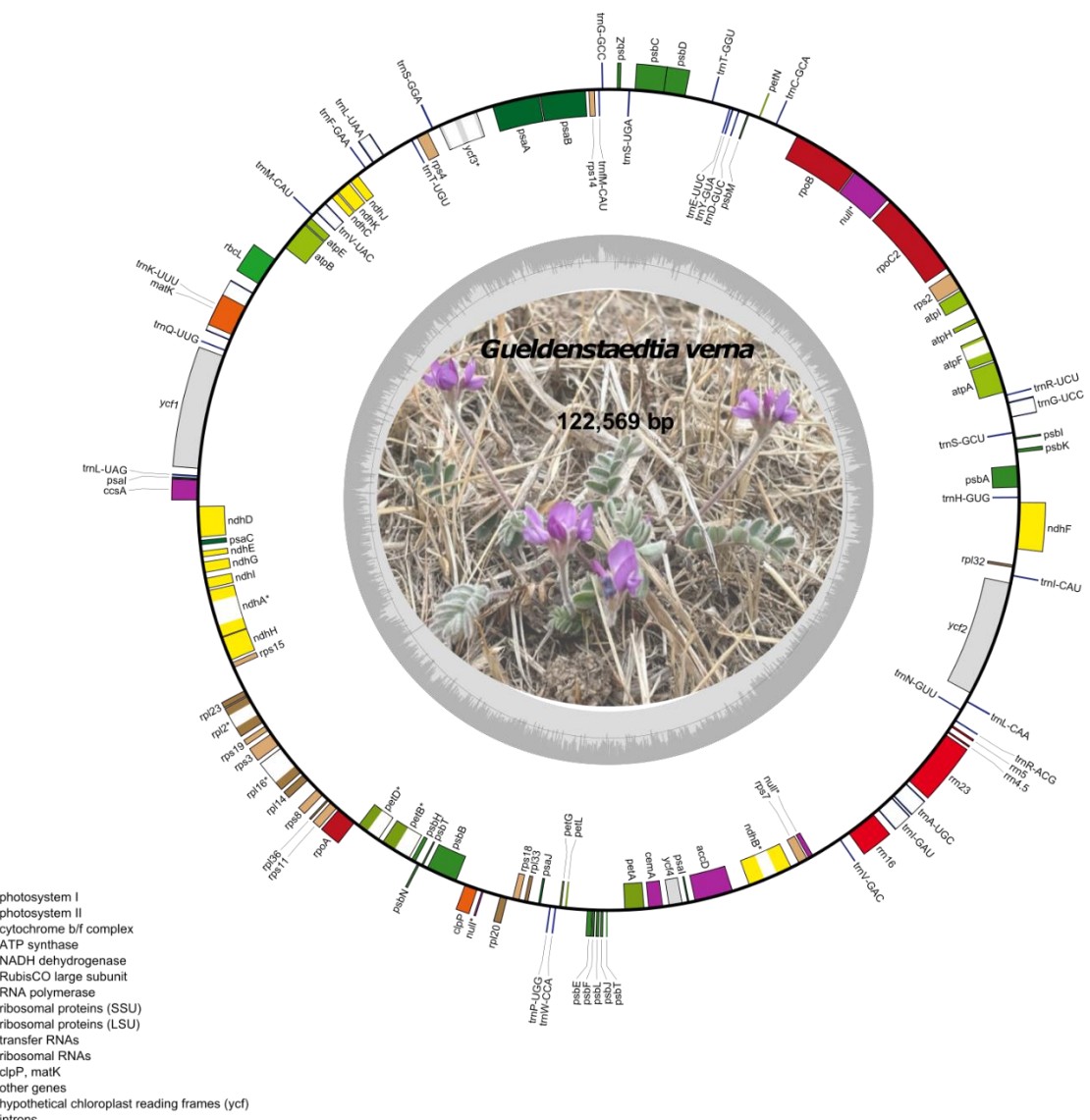

**Figure 1.** The complete chloroplast genome of *Gueldenstaedtia verna*. The genes are transcribed clockwise on the inside and counterclockwise on the outside. The darker gray in the inner circle corresponds to the GC content.

**Table 1.** Characters of 15 inverted repeat-lacking clade (IRLC) species and 1 legume species (LS, *Lotus japonicus*).

| Taxon | Genome Size (bp) | GC Contents | Gene | | | Number of Repeat [a] (F/R/C/P) [b] | Length of Total Repeats (bp) | Repeats Percentage (%) |
|---|---|---|---|---|---|---|---|---|
| | | | Coding Genes | tRNA | rRNA | | | |
| *Lotus japonicus* | 150,519 | 36.0% | 78 | 30 | 4 | 61 (26/3/2/33) | 2834 | 1.8% |
| *Glycyrrhiza lepidota* | 127,939 | 34.2% | 77 | 30 | 4 | 92 (59/4/4/25) | 4315 | 3.3% |
| *Wisteria sinensis* | 130,561 | 34.4% | 77 | 30 | 4 | 109 (70/14/0/25) | 4622 | 3.5% |
| *Wisteria brachybotrys* | 131,179 | 34.4% | 77 | 30 | 4 | 89 (50/11/2/26) | 3615 | 2.7% |
| *Astragalus mongholicus* var. *nakaianus* | 123,633 | 34.1% | 77 | 30 | 4 | 67 (37/7/1/22) | 2597 | 2.1% |

**Table 1.** *Cont.*

| Taxon | Genome Size (bp) | GC Contents | Gene | | | Number of Repeat [a] (F/R/C/P) [b] | Length of Total Repeats (bp) | Repeats Percentage (%) |
|---|---|---|---|---|---|---|---|---|
| | | | Coding Genes | tRNA | rRNA | | | |
| *Lessertia frutescens* | 122,700 | 34.2% | 77 | 30 | 4 | 35 (16/3/0/16) | 1394 | 1.1% |
| *Gueldenstaedtia verna* | 122,569 | 36.0% | 77 | 30 | 4 | 74 (51/0/0/23) | 3727 | 3.0% |
| *Tibetia himalaica* | 124,201 | 34.5% | 77 | 30 | 4 | 236 (227/1/0/8) | 11,917 | 9.5% |
| *Tibetia liangshanensis* | 122,372 | 34.7% | 77 | 30 | 4 | 93 (82/1/0/10) | 4260 | 3.4% |
| *Cicer arietinum* | 125,319 | 33.9% | 76 | 30 | 4 | 75 (45/4/1/25) | 3548 | 2.8% |
| *Medicago hybrida* | 125,208 | 33.8% | 76 | 30 | 4 | 105 (80/5/0/20) | 4062 | 3.2% |
| *Trifolium aureum* | 126,970 | 34.6% | 77 | 30 | 4 | 51 (34/3/0/14) | 2834 | 2.2% |
| *Lens culinaris* | 122,967 | 34.4% | 75 | 30 | 4 | 105 (89/0/0/16) | 4561 | 3.7% |
| *Vicia sativa* | 122,467 | 35.2% | 76 | 30 | 4 | 78 (65/0/1/12) | 6004 | 4.9% |
| *Pisum sativum* | 122,169 | 34.8% | 75 | 30 | 4 | 61 (54/1/0/6) | 2564 | 2.0% |
| *Lathyrus sativus* | 121,020 | 35.1% | 76 | 30 | 4 | 78 (50/0/0/28) | 3343 | 2.7% |

[a] Tandem repeats $\geq$ 30 bp. [b] F, forward repeat; R, reverse repeat; C, complement repeat; P, palindromic repeat.

**Table 2.** Gene loss and number of introns of 15 IRLC species and 1 legume species (LS, *Lotus japonicus*).

| | LJ | GL | WB | WS | AM | LF | GV | TH | TL | CA | MH | TA | LC | VS | PS | LS |
|---|---|---|---|---|---|---|---|---|---|---|---|---|---|---|---|---|
| | | | | | | | Gene loss | | | | | | | | | |
| *rps16* | x | x | x | x | x | x | x | x | x | x | x | x | x | x | x | x |
| *rps18* | o | o | o | o | o | o | o | o | o | o | o | o | x | o | o | o |
| *rpl23* | o | o | o | o | o | o | o | o | o | o | o | o | o | x | x | x |
| *atpE* | o | o | o | o | x | o | o | o | o | o | o | o | o | o | o | o |
| *ycf4* | o | o | o | o | o | o | o | o | o | o | x | o | x | x | x | x |
| | | | | | | | Number of introns | | | | | | | | | |
| *atpF* | 1 | 1 | 1 | 1 | 1 | 0 | 1 | 1 | 1 | 1 | 1 | 1 | 1 | 1 | 1 | 1 |
| *clpP* | 2 | 0 | 1 | 1 | 1 | 1 | 0 | 0 | 0 | 1 | 1 | 1 | 1 | 1 | 1 | 1 |
| *ndhA* | 1 | 1 | 1 | 1 | 1 | 1 | 1 | 1 | 1 | 1 | 1 | 1 | 1 | 1 | 1 | 1 |
| *ndhB* | 1 | 1 | 1 | 1 | 1 | 1 | 1 | 1 | 1 | 1 | 1 | 1 | 1 | 1 | 1 | 1 |
| *petB* | 1 | 1 | 1 | 1 | 1 | 1 | 1 | 1 | 1 | 1 | 1 | 1 | 1 | 1 | 1 | 1 |
| *petD* | 1 | 1 | 1 | 1 | 1 | 1 | 1 | 1 | 1 | 1 | 1 | 1 | 1 | 1 | 1 | 1 |
| *rpl2* | 1 | 1 | 1 | 1 | 1 | 1 | 1 | 1 | 1 | 1 | 1 | 1 | 1 | 1 | 1 | 1 |
| *rpl16* | 1 | 1 | 1 | 1 | 1 | 1 | 1 | 1 | 1 | 1 | 1 | 0 | 1 | 1 | 1 | 1 |
| *rps12* | 2 | 1 | 1 | 1 | 1 | 1 | 1 | 1 | 1 | 1 | 1 | 1 | 1 | 1 | 1 | 1 |
| *rpoC1* | 1 | 1 | 1 | 1 | 1 | 1 | 0 | 1 | 1 | 1 | 1 | 1 | 1 | 1 | 1 | 1 |
| *ycf3* | 2 | 2 | 2 | 2 | 2 | 2 | 2 | 2 | 2 | 2 | 2 | 2 | 2 | 2 | 2 | 2 |
| *trnG*-UCC | 1 | 1 | 1 | 1 | 1 | 1 | 1 | 1 | 1 | 1 | 0 | 0 | 0 | 0 | 1 | 1 |
| *trnL*-UAA | 1 | 1 | 1 | 1 | 1 | 1 | 1 | 1 | 1 | 1 | 1 | 1 | 1 | 1 | 1 | 1 |
| *trnV*-UAC | 1 | 1 | 1 | 1 | 1 | 1 | 1 | 1 | 1 | 1 | 1 | 1 | 1 | 1 | 1 | 1 |
| *trnK*-UUU | 1 | 1 | 1 | 1 | 1 | 1 | 1 | 1 | 1 | 1 | 1 | 1 | 1 | 1 | 1 | 1 |
| *trnI*-GAU | 1 | 1 | 1 | 1 | 1 | 1 | 1 | 1 | 1 | 1 | 1 | 1 | 1 | 1 | 1 | 1 |
| *trnA*-UGC | 1 | 1 | 1 | 1 | 1 | 1 | 1 | 1 | 1 | 1 | 1 | 1 | 1 | 1 | 1 | 1 |

x: loss of gene; o: intact gene; LJ, *Lotus japonicus*; GL, *Glycyrrhiza lepidota*; WB, *Wisteria brachybotrys*; WS, *Wisteria sinensis*; AM, *Astragalus mongholicus* var. *nakainus*; LF, *Lessertia frutescens*; GV, *Gueldenstaetia verna*; TH, *Tibetia himalaica*; TL, *Tibetia liangsharensis*; CA, *Cicer arietinum*; MH, *Medicago hybrida*; TA, *Trifolium aureum*; LS, *Lens culinaris*; VS, *Vicia sativa*; PS, *Pisum sativum*; LS, *Lathyrus sativus*.

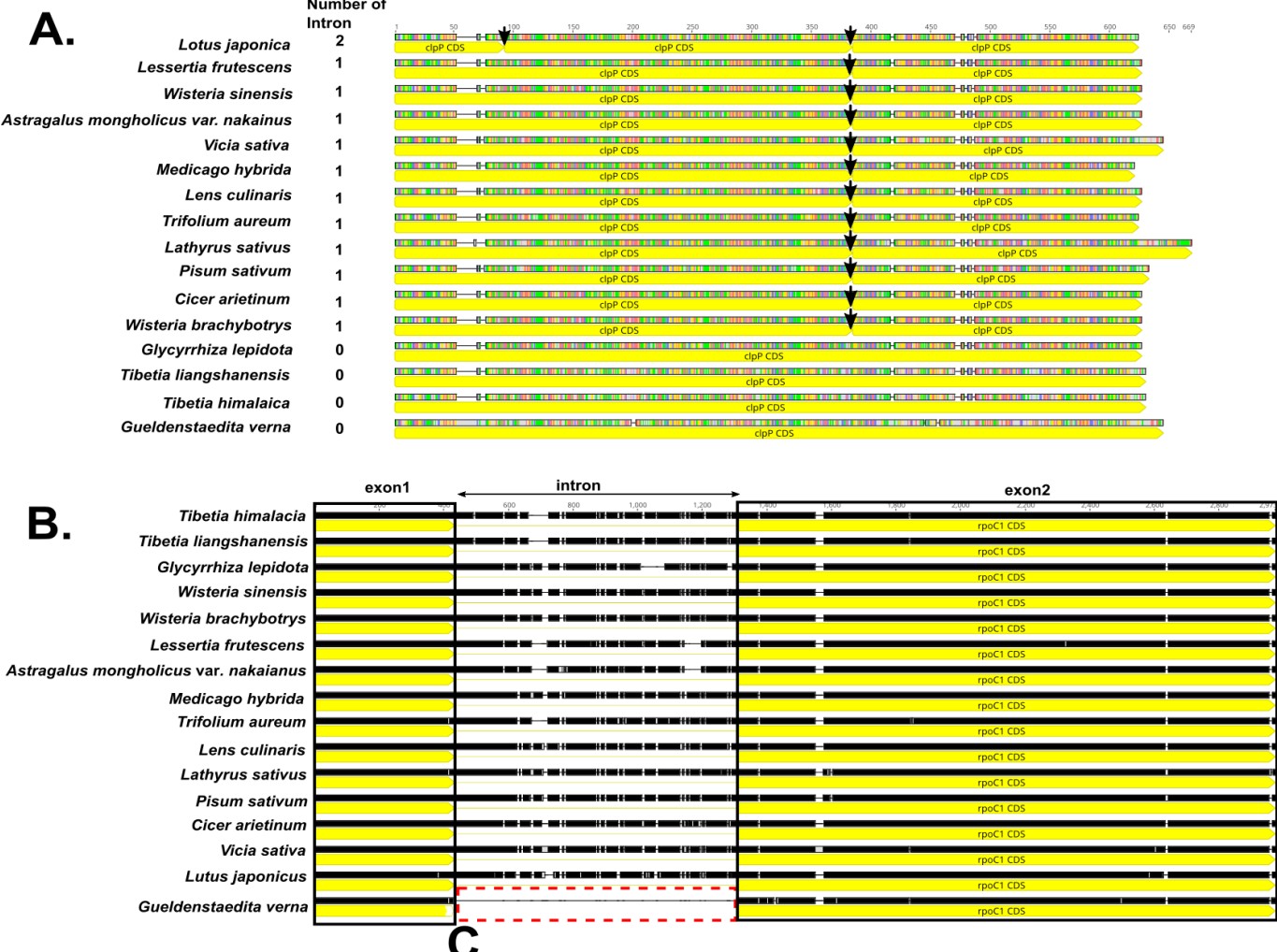

**Figure 2.** Introns of *clpP* and *rpoC1* in 19 inverted repeat-lacking clade (IRLC) species. (**A**) Introns of *clpP* in IRLC species. Arrows indicate intron locations. (**B**) Introns of *rpoC1* in IRLC species. (**C**) Loss of *rpoC1* intron in *G. verna*.

*2.3. Phylogenetic Analysis*

We conducted a maximum likelihood (ML) phylogenetic analysis based on 67 protein-coding genes from 19 species, including an outgroup (L. *japonicus*), with 67,345 bp alignment (Figure 3, Table 1). The IRLC formed a monophyletic group, subdivided into two clades: (1) *Glycyrrhiza* and *Wisteria* species and (2) Tribes Galegeae, Carganeae, Cicereae, Trifolieae, and Fabeae species. *G. verna* and genus *Tibetia* formed a single sister clade to the tribe Galegeae (L. *frutescens* and *Astragalus mongholicus* var. *nakainus*). The IRLC monophyletic group, both subclades, and the Caraganeae sister clades were well-supported by bootstrap value (100%).

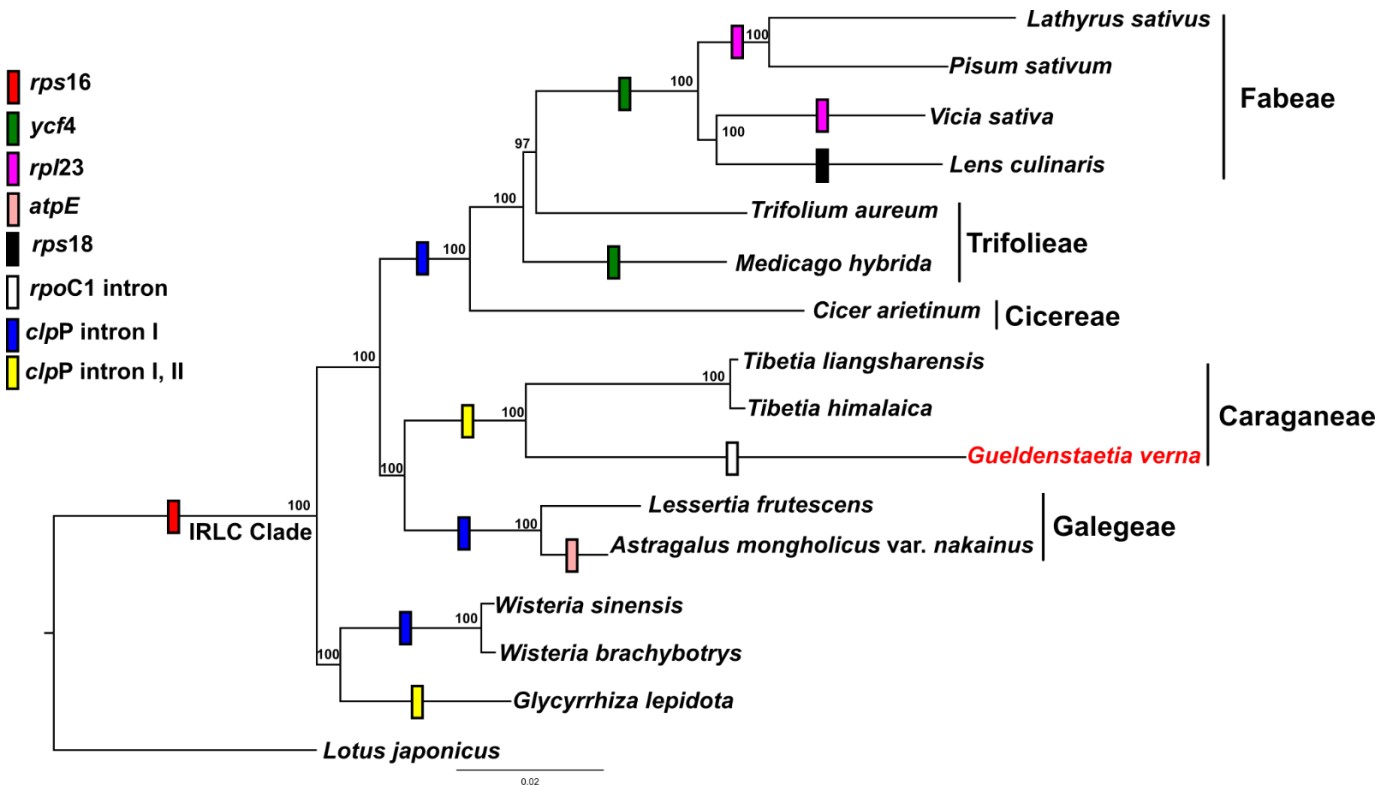

**Figure 3.** The maximum likelihood tree constructed using 67 protein-coding genes from 19 species. The colored boxes indicate loss of the gene or intron.

### 2.4. Inversion in cp of G. verna

A comparison between *G. verna* and two *Tibetia* species (*T. liangsharensis* and *T. himalaica*) detected four inversions and one reposition in *G. verna* (Figure 4). A large inversion of approximately 50 kb was located between the genes *trnK* and *psbK* (Figure 4A, Figure S2). Three inversions of approximately 28 kb, 10 kb, and 6 kb were located between *accD* and *rpl23* (Figure 4B), rps15 and *trnL* (Figure 4E), and *trnL* and *trnI* (Figure 4C), respectively. Additionally, *G. verna* showed a reposition of the *ycf1* gene (Figure 4D).

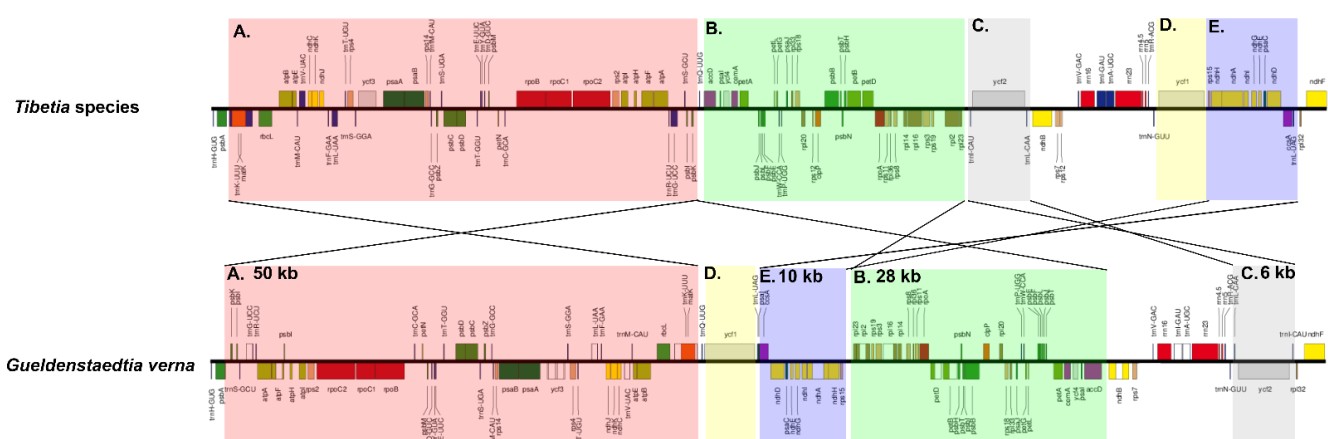

**Figure 4.** Comparison of the chloroplast genome structures of two *Tibetia* species with *G. verna*. The (**A**–**C**,**E**) boxes indicate approximately 50 kb, 28 kb, 6 kb, and 10 kb, respectively. The (**D**) box indicates the repositioning of the *ycf1* gene.

### 2.5. Substitution Analysis

We analyzed the substitution rates of 71 protein-coding genes from 18 IRLC species using *Lotus japonicus* as a reference (Figure 5 and Table S1). The median value of synonymous substitutions (*dS*) was higher than that of the non-synonymous substitutions (*dN*). The *dS* median ranged from 0.31 (*G. lepidota*) to 0.06 (*L. sativus*), and the *dN* ranged from 0.19 (*G. lepidota*) to 0.26 (*L. culinaris*). Among the analyzed genes, the highest *dN* rates were from *clpP* (0.55) and *ycf1* (0.45) in *G. verna*. The *psbI* gene had a higher *dS* rate in *L. frutescens*, *P. sativum*, and *L. sativus* than that in other species. Most of the genes' *dN/dS* values were less than 1. The exceptions were *ycf1* in *W. brachybotrys*, *W. sinensis*, T. *haimalaica*, and *T. liangsharensis*; *rps18* in *L. culianris*; and *rps7* and *ycf2* in *G. verna*.

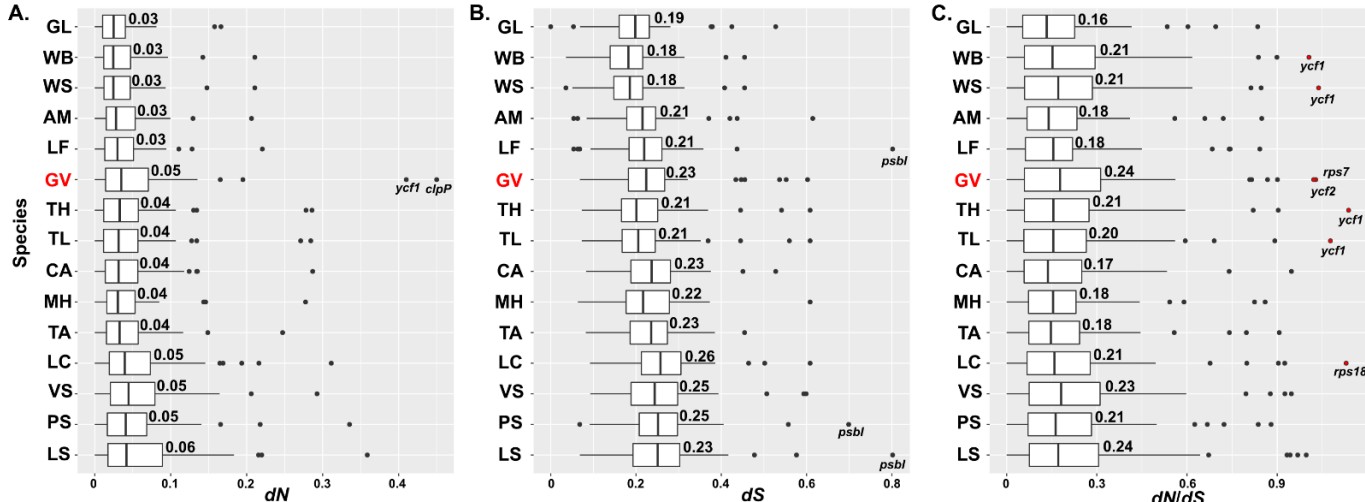

**Figure 5.** Boxplot showing the variation in non-synonymous substitutions (*dN*) (**A**), synonymous substitutions (*dS*) (**B**), and *dN/dS* (**C**) for the IRLC species. The median values are indicated above the whiskers. The red circles (**C**) show the genes of IRLC species with *dN/dS* > 1. LJ, *Lotus japonicus*; GL, *Glycyrrhiza lepidota*; WB, *Wisteria brachybotrys*; WS, *Wisteria sinensis*; AM, *Astragalus mongholicus* var. *nakainus*; LF, *Lessertia frutescens*; GV, *Gueldenstaetia verna*; TH, *Tibetia himalaica*; TL, *Tibetia liangsharensis*; CA, *Cicer arietinum*; MH, *Medicago hybrida*; TA, *Trifolium aureum*; LS, *Lens culinaris*; VS, *Vicia sativa*; PS, *Pisum sativum*; LS, *Lathyrus sativus*.

### 3. Discussion

The complete cp genome size, structure, and gene content are highly conserved in angiosperms [10]. However, rare genome characteristics, such as gene loss, inversion, and IR loss, have been reported in Fabaceae [11,21,22,31]. *Gueldenstaetia* is a genus from the Caraganaceae family, which, together with Fabaceae, belongs to the IRLC [32]. In this study, we showed the novel and complete cp genome of *G. verna* and compared it with previously reported cp genomes from related species.

Previous studies have shown that six genes (*accD*, *infA*, *rpl22*, *rps16*, *rps18*, and *ycf1*) were absent in *Trifolium subterraneum* [11], the genes *rps16* and *rpl22* were lost in *Astragalus membranaceus* [33], and the *rpl2* gene was absent in *Trifolium resupinatum* [34]. Magee et al. [22] reported that four legume species (*Glycine max*, *Trifolium subterraneum*, *Cicer arietinum*, and *Medicago truncatula*) lost the *ycf4* gene. The introns of *rps12* and *clpP* were also not found in the IRLC [22,35,36]. This study found that the *rps16* gene and two introns (*clpP* and *rpoC1*) were absent in *G. verna* (Figure 2, Table 2). Previous studies have shown that the *rps16* gene and intron 1 of *clpP* were absent from the IRLC species [37–39], except *Glycyrrhiza glabra* [37]. However, in *G. verna* and the genus *Tibetia*, the *clpP* gene lost two introns. Intron II of *clpP* has been lost independently in land plants [40], including *G. verna*, two *Tibetia* species, and *Glycyrrhiza glabra*. The loss of the *rpoC1* intron has been reported in some taxa, such as one species of *Medicago*, four species of *Passiflora*, and other

species of *Scaevola*, *Goodenia*, and Cactaceae [40,41]. This study showed that the genus *Tibetia* has an intron of *rpoC1*, which has been lost in *G. verna* (Figure 2).

The IRLC exhibits many rearrangements, such as two inversions in *Astragalus* [31], *Trifolium*, and *Vicia* [23] (Figure S2). Our results showed that *Glycyrrhiza*, *Wisteria*, *Astragalus*, *Lessertia*, and *Tibetia* had similar cp genome structures (Figure S2), whereas some variations, such as inversion and reposition, were detected in *G. verna* (using *Tibetia* as the reference, Figure 4). Hiratsuka et al. [42] and Walker et al. [43] suggested that the cp genome structure is correlated with tRNA through intermolecular recombination between tRNA sequences, while Fullerton et al. [44] reported that the G + C content affects inversion. We detected four inversions and one repositioning in *G. verna*, although its total G + C content was higher than that of other IRLC species. Our results do not support the G + C content hypotheses, and future studies are needed to describe the cp genome structure variation better. Repetitive sequences in *G. verna* are not longer or more numerous than those in other species of the IRLC (Table 1). *T. himalaica* had the highest number of repetitive sequences; however, the cp genome structure of *T. himalaica* is similar to closely related species (Figure S2). Previous studies [23,45,46] have reported that repetitive sequences are located in duplications of tRNA and flanking inversion regions in cp genomes. However, no such association was found among the repetitive sequences in *G. verna*.

Previous molecular phylogenetic studies [47] using nrDNA ITS and cpDNA *matK*, *trnL-F*, and *psbA-trnH* markers grouped *Gueldenstaedtia* and *Tibetia* into one clade. Our results revealed that *Guldenstaedtia* and *Tibetia* were in the same clade and well-supported (100% bootstrap value).

The *dS* of cpDNA is lower than that of nrDNA and higher than that of mtDNA [48]. The substitution rates of genes in the single-copy (SC) region are higher than those in the IR region [49]. Recently, many scholars [37,50–52] have attempted to solve the questions associated with genome evolution, such as structure, inversion, and rearrangement, using substitution rates. For example, Schwarz et al. [50] suggested that the *dN* and *dS* substitution rates are correlated with plastome size and rearrangements. We observed many inversions and repositionings in *G. verna*. The *dN* of *ycf1* and *accD* were higher than in other species (Figure 5A). However, *ycf1* and *accD* did not exhibit positive selection (*dN/dS* < 1). Two genes (*rps7* and *ycf2*) were positively selected with *dN/dS* > 1 (Figure 5). Our study showed that the substitution rates of *G. verna* did not support the previously reported ones [50]. In addition, localized hypermutation regions, such as *accD*, *clpP*, and *ycf1*, have been reported to accelerate substitution rates [23,24], whereas the substitution rates of the three genes in *G. verna* were not positively correlated (*dN/dS* < 1). This implies that the rate accelerations of cp genes in *G. verna* are different from those in other species, and a more comprehensive sampling of this taxon is needed to determine the evolution of cp genes in *Gueldenstaedtia*.

## 4. Materials and Methods

### 4.1. Sampling, DNA Extraction, and Sequencing

Fresh *G. verna* leaves were collected from Bolli-ri, Hwawon-eup, Dalseong-gun, Daego, Korea. The specimens were deposited at the Daegu National Science Museum. Total genomic DNA was extracted using a DNeasy Plant Mini Kit (Qiagen Inc., Valencia, CA, USA). Genomic DNA was sequenced using the Illumina HiSeq X platform (San Diego, CA, USA). We obtained 32,505,084 total reads from the 150 bp paired-end sequences with a quality value ≥Q30, accounting for 89.1%.

### 4.2. Genome Assembly, Genome Annotation, and Comparison of Genome Structure

The de novo assembly of the chloroplast genome was performed using GetOrganelle v.1.7.6.1 [53]. For coverage calculations (Figure S1), the reads were aligned using Bowtie2 [54]. The read coverage of *G. verna* is shown in Figure S1. Geseq (https://chlorobox.mpimp-golm.mpg.de/OGDraw.html, accessed on 10 November 2022) [55] was used to annotate the cp genome of *G. verna*, and tRNA gene sequences were annotated using tRNAscan-SE 2.0 [56]. Protein-coding genes and tRNAs were double-checked by identifying open reading

frames and comparing with reference genomes (Table 1) in Geneious Prime [57]. Genome mapping was performed using OrganellarGenomeDRAW (OGDRAW) (Version 1.3.1) [58], and the chloroplast genome of *G. verna* was deposited in GenBank.

We compared the cp genome of *G. verna* with published data (Table 1) of other IRLC Fabaceae species. Alignments of 16 species, including the outgroup, were found to detect genome rearrangements, such as inversion and repositioning, using Mauve v.1.1.3 on Geneious [57].

### 4.3. Repeat Analysis

The simple sequence repeats (SSRs) of *G. verna* were identified using the REPuter program [59]. Additionally, SSRs of 16 species (including the outgroup *Lotus japonicus*) were also detected (Table 1). Forward, palindromic, reverse, and complement sequences were identified with a Hamming distance of 3, minimum repeat size of 30 bp, and sequence identity $\geq$ 90%.

### 4.4. Phylogenetic Analysis

The chloroplast genome sequences of 19 taxa, including one outgroup (*Lotus japonicus*, following Xiong et al. [32] and Xia et al. [60]), were included in the phylogenetic analyses. The 67 protein-coding genes shared across taxa were extracted from each chloroplast genome and concatenated. The sequences were aligned using MAFTT [61]. The ML analysis was conducted using RAxML (version 8) [62]; the GTR + GAMMA + I model was performed using a rapid bootstrap of 1000 replications.

### 4.5. Substitution Rate Estimation

The *dN* and *dS* rates were estimated for each of the 67 cp genes using CODEML in PAML v4.8 [63]. The phylogenetic tree generated in the previous section was used as a constraint tree for all the rate comparisons. In PAML, codon frequencies were determined using the F3 $\times$ 4 model, and gapped regions were excluded with the "cleandata = 1" parameter option. The transition/transversion ratio and *dN*/*dS* values were estimated using initial values of 2.0 and 0.4, respectively.

**Supplementary Materials:** The following supporting information can be downloaded at: https://www.mdpi.com/article/10.3390/f13111942/s1, Figure S1: Coverage of *G. verna*. Whole genome resequencing reads were mapped to the assembled *G. verna*.; Figure S2: Gene rearrangement analyses among IRLC species by Mauve alignment; Table S1: Nonsynonymous (*dN*) and synonymous (*dS*) substitutions among IRLC species.

**Author Contributions:** Conceptualization, O.S. and K.S.C.; methodology, O.S.; software, O.S. and K.S.C.; validation, O.S. and K.S.C.; formal analysis, K.S.C.; investigation, O.S. and K.S.C.; resources, O.S.; data curation, O.S. and K.S.C.; writing—original draft preparation, O.S. and K.S.C.; writing—review and editing, K.S.C.; visualization, O.S. and K.S.C.; supervision, K.S.C.; project administration, O.S.; funding acquisition, O.S. All authors have read and agreed to the published version of the manuscript.

**Funding:** This research was funded by Collect and Research Native Plants on the Korean Peninsula for the Natural History exhibition of the Daegu National Science Museum (DNSM).

**Institutional Review Board Statement:** Not applicable.

**Informed Consent Statement:** Not applicable.

**Data Availability Statement:** Not applicable.

**Conflicts of Interest:** The authors declare no conflict of interest.

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
