# Peer review of "Characterization of the Chloroplast Genome Structure of Gueldenstaedtia verna (Papilionoideae) and Comparative Analyses among IRLC Species"

_forests, doi:10.3390/f13111942_

Round 1

Reviewer 1 Report

Manuscript report

' “Variations in chloroplast genome structure of Gueldenstaedtia verna (Papilionoideae) and comparative analyses among IRLC species”- First of all am skeptical about the topic using the word “Variation” and only uses one species of the kind. It’s could have been more sensible when different species of G. verna were compared.

2.      The abstract section requires some adjustments, I prefer you give precisely and concisely relating and knotting together through these: Introduction (problem statement and Justification), Methodology, results, discussion and Conclusion, as well as Implications. As at now, it’s not convincing.

3.      Introduction: Improve the section well, to show history and what’s the focus well. The story seems not to flow well. Improve the section. Having no sequence of a genome should not be a worry in science but what implications does this have on and will have in the future for this clade or genus. Check on that.

4.      In the result section, do not mix methodology with results.

5.      The result section especially at the structure characterization, give us the structure description of how this cp genome looks like and its partitioning. Clearly state the results.

6.      Explain clearly how you chose the outgroup for this study.

7.      Ensure all the scientific names follow the binomial nomenclature rule, the names are not well written here. Check on that.

8.      Improve on the discussion. This section is what makes the work look more sensible, however by just quoting other people’s work without “discussing” your results or work makes it less convincing.

9.      Improve on the English language.

10.  For any software use, please quote the version.

11.  Check on the references.

Reviewer 2 Report

Authors investigated ‘variation in chloroplast genome structure of Gueldenstaedtia verna (Papilionoideae) and comparative analyses among IRLC species.’ The concept and methodology are pretty good. The main problem is writing, there are several linguistic issues and errors; some of which are pointed out in the annotated pdf file. Authors also ignore scientific rules of writing botanical names, I find many plant names without italicized text. I also missed the robust conclusion of the study in the abstract and main manuscript as well. Some general comments are directly included in annotated pdf file.

Round 2

Reviewer 2 Report

Thank you for the revision. This version is better than the previous one, however, I still feel there are a few places to revise and there are some linguistic errors to fix as well. I recommend for language editing native speakers. The authors did not fix the name of the species 'Medicago hybrid' (page 3, line 85). Also, see the comments in the previous annotated file (page 3, line 85).  

Author Response

Thank you for your comments and english proofreading was done once more.
